# Influence of the Drying Method on the Volatile Component Profile of *Hypericum perforatum* Herb: A HS-SPME-GC/MS Study

Karolina Dudek [1,†], Marcelin Jan Pietryja [2,†], Sławomir Kurkiewicz [1,*], Małgorzata Kurkiewicz [3], Barbara Błońska-Fajfrowska [4], Sławomir Wilczyński [4] and Anna Dzierżęga-Lęcznar [1]

1 Department of Instrumental Analysis, Faculty of Pharmaceutical Sciences in Sosnowiec, Medical University of Silesia, 40-055 Katowice, Poland
2 Institute of Monastery Medicine, ul. Związkowa 20, 40-730 Katowice, Poland
3 Platomics, Jakov-Lind-Straße 15, 1020 Vienna, Austria
4 Department of Basic Biomedical Science, Faculty of Pharmaceutical Sciences in Sosnowiec, Medical University of Silesia, Kasztanowa Street 3, 41-200 Sosnowiec, Poland
* Correspondence: slawek@sum.edu.pl
† These authors contributed equally to this work.

**Abstract:** *Hypericum perforatum* L. (St. John's wort) is one of the most popular medicinal plants in the world. Due to its documented antimicrobial and antioxidant properties, it is used in the treatment of bacterial and viral infections as well as inflammations. It is also used to treat gastrointestinal diseases and mild to moderate depression. In recent years, there has been an increase in the popularity of herbal medicine. Many people collect their own herbs and dry them at home. A common choice for quick drying of fruits, vegetables and herbs at home are food dehydrator machines. There are not many publications in the scientific literature examining the quality of dried herbal material obtained in such dryers. We characterized St. John's wort harvested in southern Poland and investigated the effect of specific drying methods on the volatile component profile. The herbal raw material was dried using three methods: indoors at room temperature, in an incubator at 37 °C and in a food dehydrator machine. Volatile components were analysed by HS-SPME GC/MS. The herb dried in a food dehydrator, compared to other drying methods, retained similar or slightly smaller amounts of the compounds from the mono- and sesquiterpenes group, aromatic monoterpenes, aromatic monoterpenoids, sesquiterpenoids, aromatic sesquiterpenes and alkanes. However, monoterpenoids and compounds coming from decomposition reactions, such as alcohols, short-chain fatty acids and esters, were noticed in larger quantities. Usage of a food dehydrator at home can be a convenient alternative to drying herbs. However, due to a different profile of volatile components depending on the drying method, the amount of biologically active substances needs to be considered. By using various methods of drying, the medical effects of herbs can be enhanced or weakened; therefore, further research in this direction should be continued.

**Keywords:** drying technology; essential oil; *Hypericum perforatum*; HS-SPME/GC-MS

## 1. Introduction

*Hypericum perforatum* L. (St. John's wort) is a perennial herbaceous plant that grows naturally in meadows and forest edges. Its characteristic features are leaves with essential oil glands, which when held to the light, reveal translucent dots, giving the impression that the leaf is perforated. It is one of the most popular medicinal plants in the world. The medical use of St. John's wort has been documented by many ancient Greek physicians, herbalists and botanists such as Hippocrates, Theophrastus and Pedanius Dioscorides [1]. The herbal preparation for well-established use requires the plant to be in the form of dry extract with methanol or ethanol used as an extraction solvent. In a pharmaceutical form,

the herbal preparation for well-established use is in solid dosage forms for oral use. For traditional use, the herbal preparations are much more varied. The plant can be prepared as a powdered herbal substance, dry extract, liquid extract with different extraction solvents (suitable vegetable oils or ethanol), tincture, expressed juice or comminuted herbal substance—all of which can be represented differently in the pharmaceutical form: liquid or semi-solid dosage forms as well as herbal teas for oral use [2]. St. John's wort is used in the treatment of inflammation, bacterial and viral infections, as well as eczema, burns, neuralgia, ulcers and gastroenteritis [3]. Its use increases the feeling of satiety; therefore, it is gaining higher popularity as a curative agent against obesity-associated complications [4,5]. St John's wort exhibits antidepressant and anxiolytic properties, which is why, in the last 20 years, it has become a popular alternative to synthetic drugs in the treatment of anxiety, insomnia, depression and other psychiatric disorders [1,4,6]. It is believed that the main compounds responsible for the antidepressant effect are hypericin and hyperforin present in ethanol extracts [1,7]. St. John's wort also contains many other compounds, including flavonoids, tannins, xanthones and essential oils. Tannins and xanthones are phenolic derivatives that exhibit antifungal and anti-inflammatory effects [8]. St. John's wort flowers can contain from 7 to 12% of flavonoids. Quercetin, hyperoside, rutin and isoquercitrin are responsible for reducing oxidative stress that affects carcinogenesis or aging processes. On the other hand, flavonoid compounds, such as quercitin, kaemferol and biapigenin, due to their ability to inhibit the peroxidation of mitochondrial membranes, have gastro and neuroprotective effects [9].

St. John's wort, like any herbal medicine, can also have side effects. The volatile compounds contained in it, such as limonene or benzyl alcohol, are considered as allergens and can cause contact or inhalation allergies in sensitive people [10]. The use of St. John's wort with simultaneous exposure to sunlight carries the risk of photosensitivity, which produces changes in skin pigmentation. Hypericin is responsible for this effect. Other negative consequences that could occur are fatigue, restlessness, gastrointestinal irritations and nausea [11].

In recent years, more people care about diet and a healthy lifestyle. The healthy and organic food market is growing rapidly and is showing a strong upward trend. Along with this trend, there has been a strong increase in the popularity of herbal medicine. The medicinal plants market is developing dynamically [12,13]. According to the Market Research Future report, in 2021, this market amounted to $145 billion (USD), while, by 2030, it is expected to increase to $356 billion [14].

Food dehydrator machines used for quick drying of fruits, vegetables and herbs are gaining in popularity, the market size exceeded $2 billion (USD) in 2021. It is also expected to record more than 6% compound annual growth rate between 2022 and 2030. Many supporters of natural and herbal medicine dry herbs using home methods, including food dehydrator machines [15]. So far, the parameters of herbal dried material obtained in such dryers have not been researched in depth.

The literature data show that there is a large variation among the volatile components of plants depending on the geographical region [16,17], so it is important to study herbs from local areas. The method of drying the herb affects the amount of active ingredients, especially volatile ingredients. Therefore, the aim of the study is to characterize the volatile components of St. John's wort collected in Upper Silesia and to compare the influence of various drying methods on their profile.

## 2. Materials and Methods

### 2.1. Plant Material

The plant material in the form of *Hypericum perforatum* L. during flowering was collected on 15 July 2021 at the edge of the forest near the village of Bogacica (Stobrawa Valley, Opole Voivodeship, Upper Silesia).

### 2.2. Analysis of Volatile Compounds in St. John's Wort

The HS-SPME (headspace-solid phase microextraction) GC/MS (gas chromatography/mass spectrometry) technique was used to analyze *Hypericum perforatum*. Flowers, leaves and stems were analyzed separately. Samples of 200 mg of fresh plant were used, which corresponded to 55 mg of flowers, 58 mg of leaves or 87 mg of stems after drying. These samples were then placed in screw-cap glass vials with a capacity of 20 mL each. Those were conditioned in 50 °C for 10 min, which allowed for the volatile ingredients to move towards the gas phase. Then, the PDMS/DVB (polydimethylsiloxane/divinylbenzene) 65 μm fiber (Agilent Technologies, Inc., Santa Clara, CA, USA) was inserted into each vial kept at 50 °C, and left in contact with the headspace for the next 30 min. Subsequently, it was introduced into the split-splitless injector (a split ratio: 1:5) of the 7890A gas chromatograph (Agilent Technologies, Inc., Santa Clara, CA, USA), where the desorption of extracted compounds was carried out for 1 min in 270 °C. The GC separations were performed on a HP-5MS (5% diphenyl, 95% dimethyl polysiloxane, 60 m 0.32 mm id, 0.25 μm film thickness) fused-silica capillary column (Agilent Technologies, Inc., Santa Clara, CA, USA). Helium was used as the carrier gas at a flow rate of 2.5 mL/min. The GC oven temperature was programmed from 35 °C (isothermal for 1 min) to 240 °C at a rate of 5 °C/min. The final temperature was held for 10 min. The GC column outlet was connected directly to the ion source of the 7000 GC/MS Triple Quad mass spectrometer (Agilent Technologies, Inc., Santa Clara, CA, USA). The GC/MS interface, the ion source and the quadrupoles were kept at 270, 230 and 150 °C, respectively. The ionization energy was 70 eV. The mass spectrometer was operated in a full scan mode (*m/z* 29–450). The software used for data collection and mass spectra processing was MassHunter GC/MS Acquisition B.07.05 and MassHunter Workstation B.07.00 (Agilent Technologies, Inc., Santa Clara, CA, USA). Analyzed compounds were identified by comparison of their mass spectra with the library standards (the NIST/EPA/NIH Mass Spectral Library 2014 and the Wiley Registry of Mass Spectral Data 10th Edition) and by comparison of the Kovats retention indices (RI) calculated on HP-5MS column with the tabulated values (NIST).

### 2.3. Methods of Drying the St. John's Wort Herb

St. John's wort was dried using three methods: Method 1 (M1)—indoors in a shaded place at room temperature for 68 days; Method 2 (M2)—in an incubator (CLN 53 STD, Pol-Eko-Aparatura, Wodzisław Śląski, Poland) maintaining the temperature of 37 °C for 9 days; and Method 3 (M3)—in a food dehydrator (suszarka spożywcza typ 970.01 PP, Fabryka AGD Niewiadów, Warszawa, Poland) with a heater power of 300 W. This dryer has a fan forcing the air flow; it also maintains a constant temperature in the range of 44–49 °C. The material was dried for 24 h. The selection of the conditions, such as temperature and time, used in the respective drying methods has been chosen to obtain constant weight in order to determine the weight loss in each sample. The resulting weight loss did not depend on the drying method, only differing on the part of the plant: 72.8% for flowers, 71.3% for leaves and 57.2% for stems. After drying, the herb was stored in a sealed glass container at room temperature.

## 3. Results and Discussion

Food dehydrators grew in popularity due to some of their beneficial properties. They do not occupy much space, additionally thanks to the use of increased temperature (up to 50 °C) and air flow forced by the fan, the drying time can be shortened to 24 h. In this study, we investigated how accelerated drying affects the composition of volatile compounds in St. John's wort. We also examined if the volatile component profile differs from drying methods at lower temperatures for longer periods.

Since the herb consists of different anatomical parts (stem, leaf and flower), each of these parts was characterized separately. Table 1 shows the percentage of volatile components released from both the fresh plants and the plants dried using the three different methods mentioned earlier. All analyzed compounds, due to their chemical structure,

were divided into groups (Table 1, Figure 1). Chromatograms of volatile compounds from flowers, leaves and stems of St. John's wort dried by all methods can be visible in the Supplementary Materials (Figures S1–S3). The influence of the drying method on the percentage of the total peak area in individual groups is presented in Figure 1. When comparing different parts of the plant, the flowers showed slightly more monoterpenes and monoterpenoids than the leaves and the stems, while the stems contained slightly more aromatic sesquiterpenes. More aldehydes were retained in the leaves, especially those dried at lower temperatures (M1 and M2). This observation suggests that, although the whole herb is pharmaceutical raw material, using individual fragments of the plant selectively, we can obtain a different intensity of effects, both desirable and undesirable.

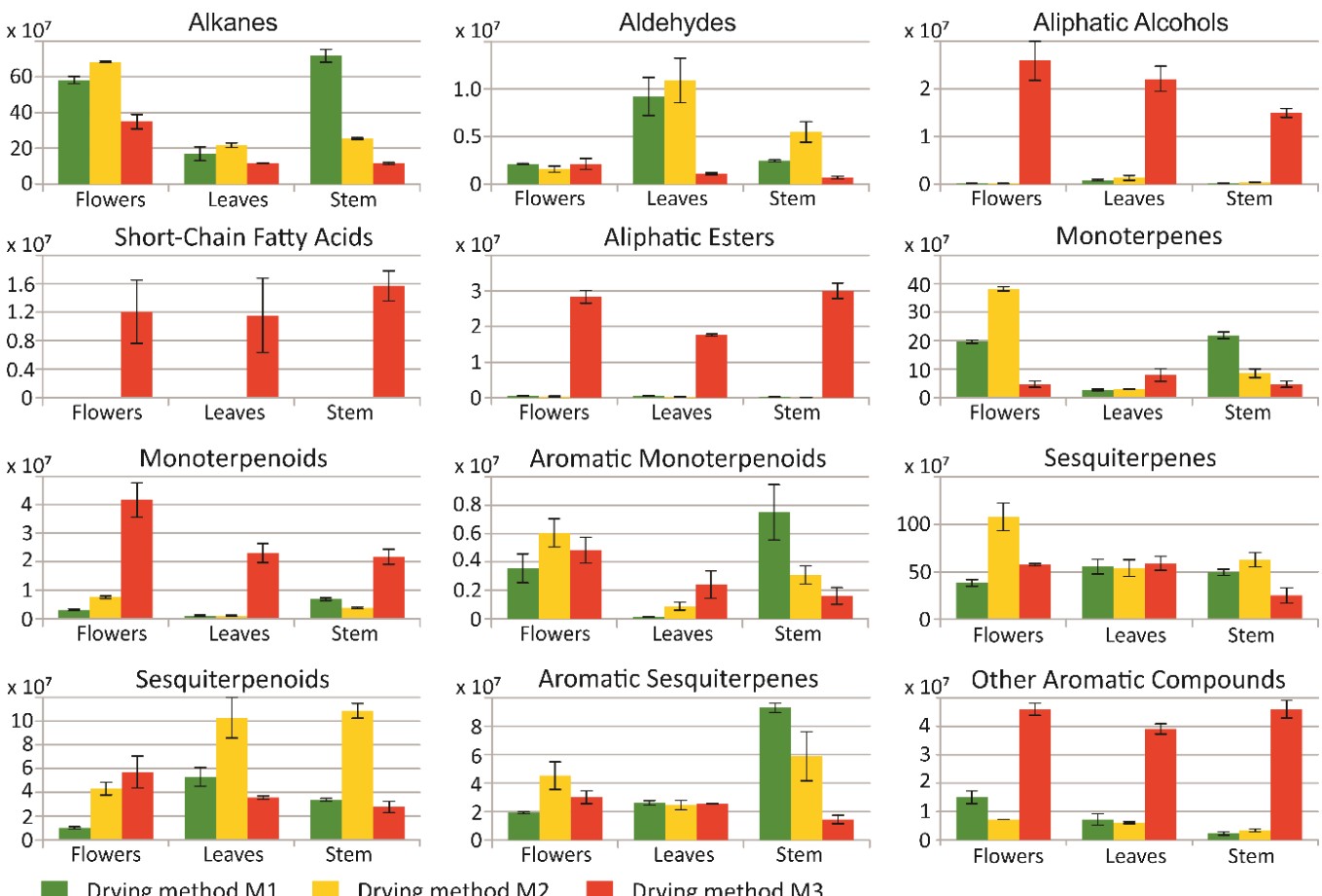

**Figure 1.** The influence of various drying methods on the volatile component profile of *Hypericum perforatum*.

Table 1. The percentage composition of the St. John's wort samples tested.

| | | | | | | Flower | | | | Leaf | | | | Stem | | | |
|---|---|---|---|---|---|---|---|---|---|---|---|---|---|---|---|---|---|
| RT [min] | Compound | Group | CAS | RI 5MS | RI NIST | Fresh | M1 | M2 | M3 | Fresh | M1 | M2 | M3 | Fresh | M1 | M2 | M3 |
| 6.10 | Propanedioic acid, dimethyl- | SCFA | 595-46-0 | 753.7 | - | - | - | - | - | - | - | - | 0.03 | - | - | - | 0.13 |
| 7.11 | Octane | Alk | 111-65-9 | 800.0 | 800.0 | 0.02 | 0.01 | 0.02 | tr | - | - | - | - | 0.05 | tr | tr | - |
| 7.11 | Hexanal | Ald | 66-25-1 | 800.0 | 800.0 | - | 0.06 | - | 0.04 | 0.28 | 0.20 | 0.08 | 0.03 | - | 0.08 | 0.22 | 0.08 |
| 8.47 | Butanoic acid, 2-methyl- | SCFA | 116-53-0 | 847.6 | 846.0 | - | - | - | 0.02 | - | - | - | 0.03 | - | - | - | 0.36 |
| 8.63 | (E)-2-Hexenal | Ald | 6728-26-3 | 853.1 | 854.0 | 0.01 | 0.03 | 0.02 | 0.04 | 1.12 | 0.42 | 0.56 | 0.06 | - | - | - | - |
| 8.70 | 3-Hexen-1-ol, (Z)- | AOH | 928-96-1 | 855.6 | 856.0 | 0.24 | 0.01 | tr | 0.02 | 1.45 | 0.08 | 0.09 | - | 0.38 | 0.01 | 0.03 | 0.06 |
| 8.95 | Octane, 2-methyl- | Alk | 3221-61-2 | 864.3 | 866.0 | 7.04 | 15.28 | 8.40 | 4.29 | 0.40 | 6.23 | 4.76 | 3.54 | 6.03 | 11.46 | 4.51 | 5.88 |
| 9.97 | Nonane | Alk | 111-84-2 | 900.0 | 900.0 | 1.54 | 3.65 | 2.09 | 1.23 | 0.41 | 1.78 | 1.10 | 1.06 | 5.37 | 3.23 | 1.66 | 1.47 |
| 10.75 | Hexanoic acid methyl ester | Est | 106-70-7 | 925.1 | 925.0 | 0.01 | 0.01 | tr | 0.25 | tr | 0.01 | tr | 0.22 | - | tr | tr | - |
| 10.84 | α-Thujene | MT | 2867-05-2 | 928.0 | 928.0 | 1.19 | 0.26 | 0.15 | 0.15 | 1.94 | 0.44 | 0.33 | 0.10 | 2.13 | 1.55 | 0.60 | 0.10 |
| 11.06 | α-Pinene | MT | 80-56-8 | 935.0 | 935.0 | 7.42 | 8.41 | 7.47 | 2.60 | 3.09 | 1.07 | 0.71 | 1.95 | 5.19 | 2.26 | 2.36 | 1.57 |
| 11.60 | Isocaproic acid | SCFA | 646-07-1 | 952.4 | 955.0 | - | - | - | 0.35 | - | - | - | 0.48 | - | - | - | 0.75 |
| 11.87 | Benzaldehyde | Ar | 100-52-7 | 961.1 | 961.0 | 0.01 | 0.04 | 0.02 | 1.32 | 0.01 | 0.22 | 0.07 | 0.92 | 0.03 | 0.04 | 0.06 | 2.56 |
| 12.19 | Nonane, 3-methyl- | Alk | 5911-04-6 | 971.4 | 976.0 | 1.43 | 5.33 | 3.05 | 2.34 | 0.12 | 2.09 | 1.47 | 2.34 | 3.76 | 4.87 | 2.28 | 2.31 |
| 12.30 | Sabinene | MT | 3387-41-5 | 974.9 | 975.0 | 1.30 | 0.09 | 0.06 | tr | 7.77 | 0.28 | 0.17 | 0.01 | 1.31 | 0.10 | 0.15 | 0.03 |
| 12.42 | β-Pinene | MT | 127-91-3 | 978.8 | 979.0 | 1.95 | 1.77 | 2.39 | 1.33 | 3.64 | 1.11 | 0.69 | 1.17 | 4.36 | 2.47 | 1.37 | 1.23 |
| 12.58 | Hexanoic acid | SCFA | 142-62-1 | 983.9 | 984.0 | - | - | - | 0.67 | - | - | - | 0.47 | - | - | - | 1.09 |
| 12.71 | 6-Methyl-5-heptene-2-one | Ket | 110-93-0 | 988.1 | 988.0 | 0.19 | 0.11 | 0.08 | 0.04 | 0.01 | 0.30 | 0.11 | 0.05 | 0.02 | 0.05 | 0.23 | 0.12 |
| 12.83 | β-myrcene | MT | 123-35-3 | 992.0 | 992.0 | 1.57 | 1.49 | 1.31 | 0.25 | 3.38 | 0.35 | 0.38 | 0.16 | 1.88 | 2.12 | 0.79 | 0.06 |
| 13.08 | Decane | Alk | 124-18-5 | 1000.0 | 1000.0 | 0.25 | 0.37 | 0.18 | 0.08 | 0.21 | 0.12 | 0.08 | 0.29 | 0.27 | 0.18 | 0.12 | 0.36 |
| 13.28 | α-Phellandrene | MT | 99-83-2 | 1006.5 | 1006.0 | 0.12 | 0.24 | 0.26 | 0.01 | 0.33 | 0.05 | 0.12 | 0.05 | 0.22 | 0.31 | 0.19 | 0.06 |
| 13.44 | 2,4-Heptadienal, (E,E)- | Ald | 4313-03-5 | 1011.6 | 1012.0 | - | 0.03 | - | - | - | 0.18 | - | - | - | 0.03 | 0.05 | - |
| 13.51 | Acetic acid, hexyl ester | Est | 142-92-7 | 1013.9 | 1014.0 | 0.16 | - | - | 0.01 | 0.01 | - | - | tr | - | - | - | - |
| 13.65 | α-Terpinene | MT | 99-86-5 | 1018.4 | 1018.0 | 0.77 | 0.51 | 0.43 | 0.10 | 1.40 | 0.19 | 0.19 | 0.04 | 1.20 | 0.87 | 0.49 | 0.05 |
| 13.91 | p-Cymene | ArMT | 99-87-6 | 1026.8 | 1027.0 | 0.72 | 1.75 | 0.96 | 0.92 | 0.33 | 1.48 | 1.39 | 0.86 | 1.22 | 4.90 | 2.87 | 0.54 |
| 14.03 | 2-Octanol, 2-methyl- | AOH | 628-44-4 | 1030.6 | | - | - | - | - | 2.65 | - | - | 2.79 | - | - | - | 3.60 |
| 14.05 | Limonene | MT | 138-86-3 | 1031.3 | 1031.0 | 1.58 | 1.26 | 1.20 | 0.01 | 3.20 | 0.45 | 0.47 | 0.03 | 2.29 | 1.78 | 0.84 | - |
| 14.17 | Benzyl alcohol | Ar | 100-51-6 | 1035.2 | 1035.6 | 0.28 | 0.12 | 0.06 | 0.05 | 0.03 | 0.06 | 0.06 | 0.02 | 0.14 | 0.01 | 0.01 | 0.09 |
| 14.29 | β-trans-ocimene | MT | 3779-61-1 | 1039.0 | 1039.0 | 2.28 | 0.70 | 0.75 | 0.13 | 6.00 | 0.07 | 0.12 | 0.12 | 0.82 | 0.29 | 0.16 | 0.03 |
| 14.49 | Benzenacetaldehyd | Ar | 122-78-1 | 1045.5 | 1045.6 | 0.03 | 0.03 | 0.01 | 0.84 | 0.05 | 0.13 | 0.05 | 0.52 | 0.47 | 0.01 | 0.04 | 1.34 |
| 14.64 | β-cis-Ocimene | MT | 3338-55-4 | 1050.3 | 1051.0 | 14.96 | 2.17 | 2.22 | 0.33 | 25.23 | 0.19 | 0.29 | 0.31 | 3.20 | 0.82 | 0.39 | 0.15 |
| 14.98 | γ-Terpinene | MT | 99-85-4 | 1061.3 | 1061.4 | 1.59 | 0.85 | 0.77 | 0.26 | 2.56 | 0.28 | 0.22 | 0.25 | 2.62 | 2.67 | 0.97 | 0.08 |
| 15.09 | Decane, 2-methyl- | Alk | 6975-98-0 | 1064.8 | 1065.0 | 5.55 | 9.13 | 5.21 | 5.81 | 0.18 | 3.48 | 3.98 | 5.41 | 3.36 | 10.71 | 4.16 | 3.41 |

**Table 1.** *Cont.*

| RT [min] | Compound | Group | CAS | RI 5MS | RI NIST | Flower Fresh | M1 | M2 | M3 | Leaf Fresh | M1 | M2 | M3 | Stem Fresh | M1 | M2 | M3 |
|---|---|---|---|---|---|---|---|---|---|---|---|---|---|---|---|---|---|
| 15.42 | Linalool oxide | MTO | 5989-33-3 | 1075.5 | 1075.0 | tr | 0.03 | 0.02 | 0.70 | tr | 0.02 | 0.01 | 0.64 | 0.01 | 0.01 | 0.02 | 1.23 |
| 15.90 | α-Terpinolen | MT | 586-62-9 | 1091.0 | 1091.0 | 0.41 | 0.26 | 0.29 | 0.35 | 0.92 | 0.05 | 0.07 | 0.24 | 0.70 | 0.50 | 0.20 | 0.35 |
| 16.18 | Undecane | Alk | 1120-21-4 | 1100.0 | 1100.0 | 2.09 | 4.17 | 3.38 | 3.99 | 0.48 | 3.28 | 2.27 | 4.13 | 7.57 | 6.81 | 3.97 | 2.69 |
| 16.63 | Benzeneethanol | Ar | 60-12-8 | 1115.2 | 1115.1 | 0.76 | 0.19 | 0.07 | 1.12 | 0.09 | 0.19 | 0.11 | 0.91 | 1.08 | 0.01 | 0.04 | 1.76 |
| 17.09 | Neo-allo-ocimene | MT | 7216-56-0 | 1130.6 | 1131.0 | 0.83 | 0.41 | 0.51 | 0.05 | 1.99 | 0.03 | 0.05 | 0.05 | 0.22 | 0.21 | 0.15 | 0.02 |
| 17.14 | 1,3,8-p-Menthatriene | MT | 460-01-5 | 1132.3 | 1132.0 | 0.22 | 0.34 | 0.40 | 0.03 | 0.19 | 0.03 | 0.05 | 0.15 | 0.10 | 0.22 | 0.18 | 0.15 |
| 17.47 | Alloocimene | MT | 673-84-7 | 1143.4 | 1142.0 | 0.44 | 0.44 | 0.55 | 0.23 | 0.16 | 0.04 | 0.06 | 0.11 | 0.03 | 0.22 | 0.15 | 0.29 |
| 17.63 | Verbenol | MTO | 473-67-6 | 1148.8 | 1148.0 | - | tr | tr | 0.18 | - | tr | tr | 0.24 | tr | tr | tr | 0.34 |
| 18.16 | Acetic acid, phenylmethyl ester | Ar | 140-11-4 | 1166.7 | 1165.0 | 0.19 | - | - | - | tr | - | - | - | - | - | - | - |
| 18.29 | Undecane, 3-methyl- | Alk | 1002-43-3 | 1171.0 | 1171.0 | 0.16 | 0.22 | 0.21 | 0.33 | 0.01 | 0.13 | 0.11 | 0.23 | 0.18 | 0.29 | 0.19 | 0.23 |
| 18.38 | Benzoic acid, ethyl ester | Ar | 93-89-0 | 1174.1 | 1172.9 | 0.01 | 0.11 | 0.01 | 0.02 | 0.03 | tr | 0.01 | 0.03 | tr | - | - | 0.04 |
| 18.46 | trans-Linalool 3,7-oxide | MTO | 39028-58-5 | 1176.8 | 1178.0 | - | tr | tr | 0.15 | - | tr | tr | 0.20 | - | - | tr | 0.33 |
| 18.61 | Terpinen-4-ol | MTO | 562-74-3 | 1181.8 | 1181.5 | 0.18 | 0.16 | 0.25 | 0.99 | 0.87 | 0.13 | 0.13 | 1.00 | 0.13 | 0.39 | 0.30 | 1.27 |
| 18.80 | p-Cymen-8-ol | ArMTO | 1197-01-9 | 1188.2 | 1188.0 | tr | 0.01 | 0.01 | 0.19 | tr | tr | 0.01 | 0.23 | 0.03 | 0.02 | 0.10 | 0.28 |
| 18.99 | α-Terpineol | MTO | 98-55-5 | 1194.6 | 1195.0 | tr | 0.04 | 0.10 | 0.50 | 0.02 | 0.02 | 0.01 | 0.47 | tr | 0.01 | 0.01 | 0.53 |
| 19.10 | Methyl salicylate | Ar | 119-36-8 | 1198.3 | 1198.0 | 0.32 | 0.12 | 0.01 | - | 0.11 | 0.01 | 0.02 | - | tr | tr | 0.01 | - |
| 19.15 | Dodecane | Alk | 112-40-3 | 1200.0 | 1200.0 | 0.02 | 0.05 | 0.07 | 0.04 | tr | 0.02 | 0.04 | 0.01 | 0.02 | 0.06 | 0.06 | 0.02 |
| 19.33 | Decanal | Ald | 112-31-2 | 1206.4 | 1206.0 | 0.02 | 0.04 | 0.04 | 0.06 | 0.01 | 0.04 | 0.07 | 0.01 | 0.03 | 0.01 | 0.08 | 0.03 |
| 19.60 | Verbenone | MTO | 80-57-9 | 1216.0 | 1217.0 | 0.02 | 0.02 | tr | 1.56 | tr | 0.01 | 0.01 | 1.39 | 0.02 | tr | - | 1.91 |
| 20.07 | Methyl thymyl ether | ArMTO | 1076-56-8 | 1232.6 | 1233.0 | 0.11 | 0.18 | 0.17 | 0.06 | tr | 0.02 | 0.07 | 0.07 | 0.01 | 0.34 | 0.12 | 0.05 |
| 20.21 | Carvacrol methyl ether | ArMTO | 6379-73-3 | 1237.6 | 1238.0 | 0.38 | 0.13 | 0.11 | 0.03 | 0.56 | 0.01 | 0.01 | 0.05 | 0.10 | 0.16 | 0.09 | 0.03 |
| 20.84 | β-Phenethyl acetate | ArMTO | 103-45-7 | 1259.9 | 1260.0 | 1.45 | - | - | 0.12 | 0.02 | tr | - | 0.07 | tr | - | - | 0.07 |
| 20.84 | 3-Carvomenthenone | MTO | 89-81-6 | 1259.9 | 1259.0 | - | 0.03 | 0.03 | - | 0.03 | tr | tr | - | - | 0.01 | tr | - |
| 20.97 | Dodecane, 2-methyl- | Alk | 1560-97-0 | 1264.5 | 1265.0 | 5.33 | 5.62 | 5.44 | 7.00 | 0.01 | 1.68 | 3.81 | 3.12 | 0.90 | 4.15 | 3.75 | 2.35 |
| 21.97 | Tridecane | Alk | 629-50-5 | 1300.0 | 1300.0 | 0.91 | 1.34 | 1.44 | 1.65 | 0.01 | 0.88 | 1.01 | 0.82 | 1.21 | 1.48 | 1.50 | 1.38 |
| 22.64 | Decanoic acid, methyl ester | Est | 110-42-9 | 1325.2 | 1325.0 | - | tr | tr | 0.22 | - | - | - | 0.17 | - | tr | tr | 0.25 |
| 22.84 | Benzoic acid, 2-methylpropyl ester | Ar | 120-50-3 | 1332.7 | 1331.2 | 0.01 | 0.48 | 0.06 | 0.01 | tr | 0.02 | 0.02 | 0.03 | tr | 0.04 | 0.02 | tr |
| 23.09 | Benzoic acid, 2-methoxy-, methyl ester | Ar | 606-45-1 | 1342.1 | 1335.0 | tr | 0.04 | tr | tr | 0.01 | tr | 0.01 | tr | 0.01 | tr | tr | tr |
| 23.14 | Elemene isomer | ST | - | 1344.0 | 1343.7 | 0.04 | 0.01 | 0.10 | 0.09 | 0.35 | 0.05 | 0.11 | 0.16 | 0.11 | tr | 0.03 | 0.10 |
| 23.47 | α-Cubebene | ST | 17699-14-8 | 1356.4 | 1356.0 | 0.22 | 0.25 | 1.10 | 0.58 | 0.34 | 1.40 | 1.42 | 0.95 | 0.15 | 0.20 | 1.06 | 0.62 |
| 23.57 | α-Longipinene | ST | 5989-08-2 | 1360.2 | 1356.4 | 0.36 | 0.49 | 0.54 | 0.71 | 0.11 | 0.33 | 0.26 | 0.58 | 0.50 | 0.59 | 0.89 | 0.58 |

**Table 1.** *Cont.*

| | | | | | | Flower | | | | Leaf | | | | Stem | | | |
|---|---|---|---|---|---|---|---|---|---|---|---|---|---|---|---|---|---|
| RT [min] | Compound | Group | CAS | RI 5MS | RI NIST | Fresh | M1 | M2 | M3 | Fresh | M1 | M2 | M3 | Fresh | M1 | M2 | M3 |
| 24.09 | Ylangene | ST | 14912-44-8 | 1379.7 | 1377.0 | 0.39 | 0.60 | 0.91 | 0.87 | 0.20 | 1.07 | 0.90 | 0.97 | 0.43 | 0.69 | 1.16 | 0.71 |
| 24.21 | α-Copaene | ST | 3856-25-5 | 1384.2 | 1384.0 | 0.34 | 0.62 | 1.41 | 1.34 | 0.48 | 1.80 | 2.06 | 1.90 | 0.19 | 0.40 | 1.54 | 1.22 |
| 24.36 | α-Cedrene | ST | 469-61-4 | 1389.8 | 1386.0 | tr | 0.02 | 0.31 | 0.04 | 0.01 | 0.51 | 0.17 | 0.08 | tr | 0.30 | 0.17 | 0.04 |
| 24.48 | β-Bourbonene | ST | 5208-59-3 | 1394.4 | 1394.5 | 0.08 | 0.15 | 0.50 | 0.42 | 0.31 | 1.53 | 1.01 | 0.89 | 0.18 | 0.62 | 1.40 | 0.75 |
| 25.05 | β-cis-Caryophyllene | ST | 118-65-0 | 1416.7 | 1413.4 | 0.65 | 0.16 | 0.10 | 0.06 | tr | 0.02 | 0.06 | 0.04 | 0.03 | 0.08 | 0.08 | 0.08 |
| 25.12 | α-Gerjunene | ST | 489-40-7 | 1419.4 | 1419.0 | 0.10 | 0.10 | 0.16 | 0.19 | 0.21 | 0.10 | 0.42 | 0.37 | 0.06 | tr | 0.20 | 0.20 |
| 25.28 | Cedrene | ST | 11028-42-5 | 1425.8 | 1422.0 | 1.13 | 1.00 | 0.48 | 0.61 | 0.69 | 5.51 | 2.75 | 2.18 | 2.00 | 2.88 | 2.35 | 2.03 |
| 25.41 | β-Caryophyllen | ST | 87-44-5 | 1431.0 | 1431.0 | 12.73 | 11.52 | 11.62 | 10.87 | 6.81 | 5.26 | 12.46 | 13.42 | 6.37 | 7.86 | 10.44 | 12.49 |
| 25.61 | β-Copaene | ST | 18252-44-3 | 1438.9 | 1436.9 | 0.27 | 0.39 | 1.08 | 1.01 | 0.55 | 1.62 | 2.20 | 1.69 | 0.23 | 0.34 | 1.22 | 1.23 |
| 25.89 | Aromandendrene | ST | 489-39-4 | 1450.0 | 1440.0 | 0.18 | 0.30 | 0.75 | 0.77 | 0.30 | 1.66 | 0.93 | 0.71 | 0.17 | 0.45 | 0.69 | 0.57 |
| 26.14 | (E)-β-Famesene | ST | 18794-84-8 | 1459.9 | 1460.0 | 4.63 | 3.86 | 4.73 | 3.99 | 0.58 | 7.87 | 4.40 | 3.34 | 11.73 | 1.30 | 9.73 | 2.82 |
| 26.17 | α-Himachalene | ST | 3853-83-6 | 1461.1 | 1460.6 | 0.85 | 1.52 | 1.26 | 1.66 | 0.13 | 0.87 | 0.71 | 1.53 | 0.84 | 3.08 | 1.88 | 1.45 |
| 26.26 | α-Humulene | ST | 6753-98-6 | 1464.7 | 1464.5 | 0.53 | 0.52 | 1.02 | 0.92 | 0.63 | 1.32 | 1.85 | 1.23 | 0.40 | 0.57 | 1.16 | 0.99 |
| 26.56 | β-Acoradiene | ST | 43219-80-3 | 1476.6 | 1471.0 | 0.05 | 0.13 | 0.63 | 0.26 | 0.12 | 1.94 | 1.15 | 0.63 | 0.07 | 1.51 | 0.85 | 0.42 |
| 26.78 | γ-Muurolene | ST | 30021-74-0 | 1485.3 | 1485.2 | 2.18 | 1.85 | 3.91 | 3.97 | 1.71 | 8.08 | 5.70 | 4.46 | 3.71 | 2.29 | 4.56 | 3.65 |
| 26.87 | α-Curcumene | ArST | 644-30-4 | 1488.9 | 1488.0 | 0.30 | 0.84 | 0.92 | 1.01 | 0.10 | 1.41 | 0.80 | 0.85 | 0.83 | 3.82 | 3.30 | 1.04 |
| 26.91 | Germacrene D | ST | 23986-74-5 | 1490.5 | 1491.0 | 0.69 | 0.29 | 1.77 | 1.35 | 7.53 | 2.21 | 5.38 | 3.11 | 1.80 | 0.03 | 1.39 | 1.26 |
| 27.01 | Alloaromadendrene | ST | 25246-27-9 | 1494.4 | 1487.0 | 4.01 | 1.08 | 2.84 | 1.88 | 1.47 | 2.40 | 3.24 | 2.94 | 5.86 | 1.01 | 1.91 | 2.33 |
| 27.23 | γ-Amorphene | ST | 6980-46-7 | 1503.4 | 1495.0 | 0.31 | 0.29 | 1.12 | 1.03 | 0.84 | 1.72 | 1.85 | 1.33 | 0.38 | 0.25 | 0.90 | 0.94 |
| 27.34 | α-Muurolene | ST | 10208-80-7 | 1508.0 | 1507.3 | 0.21 | 0.28 | 0.86 | 1.11 | 0.88 | 1.61 | 1.58 | 1.48 | 0.25 | 0.25 | 0.83 | 1.28 |
| 27.43 | β-Himachalene | ST | 1461-03-6 | 1511.8 | 1511.8 | 1.02 | 0.94 | 1.12 | 1.12 | 0.48 | 0.41 | 0.37 | 0.62 | 1.88 | 0.79 | 1.26 | 0.44 |
| 27.71 | γ-Cadinene | ST | 39029-41-9 | 1523.5 | 1524.0 | 0.80 | 0.96 | 2.44 | 3.41 | 1.69 | 4.97 | 4.39 | 3.70 | 1.06 | 1.12 | 2.89 | 3.09 |
| 27.91 | δ-Cadinene | ST | 483-76-1 | 1531.9 | 1530.8 | 1.37 | 1.86 | 4.37 | 5.31 | 3.43 | 8.74 | 6.95 | 5.92 | 1.63 | 1.48 | 3.97 | 4.07 |
| 28.27 | α-Cadinene | ST | 24406-05-1 | 1547.1 | 1544.0 | 0.18 | 0.22 | 0.71 | 0.83 | 0.57 | 1.26 | 1.25 | 0.97 | 0.27 | 0.25 | 0.64 | 0.67 |
| 28.42 | α-Calacorene | ArST | 21391-99-1 | 1553.4 | 1550.0 | 0.03 | 0.15 | 0.34 | 0.56 | 0.02 | 0.69 | 0.74 | 0.64 | 0.01 | 0.14 | 0.44 | 0.42 |
| 28.74 | Nerolidol | STO | 7212-44-4 | 1566.8 | 1566.0 | 0.05 | 0.10 | 0.22 | 0.16 | 0.03 | 0.43 | 0.45 | 0.10 | 0.01 | 0.01 | 0.14 | 0.08 |
| 28.97 | 3-Hexen-1-ol, benzoate, (Z)- | Ar | 25152-85-6 | 1576.5 | 1571.0 | 0.08 | 0.05 | 0.04 | - | 0.28 | 0.02 | 0.11 | - | tr | tr | 0.03 | - |
| 29.27 | Spathulenol | STO | 6750-60-3 | 1589.1 | 1590.0 | - | 0.17 | 0.65 | 1.00 | 0.03 | 2.19 | 2.95 | 0.91 | 0.01 | 0.31 | 1.79 | 1.04 |
| 29.44 | Caryophyllene oxide | STO | 1139-30-6 | 1596.2 | 1596.0 | 0.07 | 0.31 | 0.56 | 1.83 | 0.09 | 1.29 | 2.04 | 1.49 | 0.04 | 1.12 | 3.83 | 1.77 |
| 29.68 | Mintketone | STO | 73809-82-2 | 1606.7 | 1603.0 | - | 0.05 | 0.18 | 0.01 | tr | 1.12 | 1.40 | 0.05 | tr | 0.15 | 0.86 | 0.04 |
| 29.90 | Ledol | STO | 577-27-5 | 1616.4 | 1616.0 | 0.03 | 0.04 | 0.07 | 0.34 | 0.17 | 0.07 | 0.54 | 0.35 | 0.04 | tr | 0.29 | 0.41 |
| 31.00 | α-Cardinol | STO | 481-34-5 | 1665.3 | 1665.4 | 0.01 | 0.01 | tr | 0.44 | 0.18 | tr | tr | 0.34 | 0.07 | tr | tr | 0.45 |
| 31.47 | Cadalene | ArST | 483-78-3 | 1686.2 | 1688.1 | tr | 0.25 | 0.28 | 0.36 | tr | 0.40 | 0.42 | 0.36 | tr | 0.30 | 0.37 | 0.40 |

**Table 1.** *Cont.*

| | | | | | | Flower | | | | Leaf | | | | Stem | | | |
|---|---|---|---|---|---|---|---|---|---|---|---|---|---|---|---|---|---|
| RT [min] | Compound | Group | CAS | RI 5MS | RI NIST | Fresh | M1 | M2 | M3 | Fresh | M1 | M2 | M3 | Fresh | M1 | M2 | M3 |
| 31.78 | Heptadecane | Alk | 629-78-7 | 1700.0 | 1700.0 | 0.01 | 0.23 | 0.37 | 0.76 | 0.02 | 0.15 | 1.42 | 0.41 | 0.01 | 0.06 | 0.21 | 0.33 |
| 32.32 | Methyl tetradecanoate | Est | 124-10-7 | 1725.4 | 1726.0 | - | tr | tr | 0.31 | - | - | - | 0.25 | - | tr | tr | 0.54 |
| 33.37 | Benzyl benzoate | Ar | 120-51-4 | 1774.6 | 1774.0 | tr | 0.03 | 0.02 | tr | tr | 0.01 | 0.02 | tr | tr | tr | tr | 0.02 | 0.03 |
| 33.91 | Octadecane | Alk | 593-45-3 | 1800.0 | 1800.0 | tr | 0.06 | 0.02 | 0.09 | tr | 0.04 | 0.02 | 0.03 | tr | 0.01 | 0.01 | 0.03 |
| 35.95 | Nonadecane | Alk | 629-92-5 | 1900.0 | 1900.0 | tr | 0.18 | 0.18 | 0.49 | tr | 0.02 | 0.04 | 0.17 | tr | 0.01 | 0.02 | 0.07 |
| 36.48 | Hexadecanoic acid, methyl ester | Est | 112-39-0 | 1927.2 | 1927.3 | - | tr | tr | 1.03 | - | tr | tr | 0.81 | - | tr | tr | 2.76 |
| 37.80 | Hexadecanoic acid, ethyl ester | Est | 628-97-7 | 1994.9 | 1995.7 | - | - | - | 0.28 | - | - | - | 0.12 | - | - | - | 0.76 |
| 39.73 | Linoleic acid, methyl ester | Est | 112-63-0 | 2098.4 | 2099.0 | - | - | - | 0.43 | - | - | - | 0.07 | - | - | - | 1.18 |
| 39.82 | Oleic acid, methyl ester | Est | 112-62-9 | 2103.4 | 2103.0 | - | - | - | 0.58 | - | - | - | 0.15 | - | - | - | 1.19 |
| 40.26 | Octadecanoic acid, methyl ester | Est | 112-61-8 | 2127.9 | 2128.0 | - | - | - | 0.06 | - | - | - | - | - | - | - | 0.07 |
| 40.92 | Linoleic acid ethyl ester | Est | 544-35-4 | 2164.8 | 2162.9 | - | - | - | 0.06 | - | - | - | - | - | - | - | 0.13 |
| 41.04 | Oleic acid, ethyl ester | Est | 111-62-6 | 2171.5 | 2171.0 | - | - | - | 0.06 | - | - | - | - | - | - | - | 0.12 |

RI 5MS—Kovats retention index; Calculated on the HP-5MS column. RI NIST—Kovats retention index from the NIST and Wiley library, CAS—Chemical Abstracts Service Registry Number, tr: trace (<0.01%). Groups: Alk-alkanes, AOH—aliphatic alcohols, ALD—aldehydes, SCFA—short-chain fatty acids, Ket—aliphatic ketones, Est—aliphatic esters, MT—monoterpenes, MTO—monoterpenoids, ArMT—aromatic monoterpenes, ArMTO—aromatic monoterpenoids, ST—sesquiterpenes, STO—sesquiterpenoids, ArST—aromatic sesquiterpenes, Ar—other aromatic compounds.

Although microscopic examinations of the dried material have not been performed in this study, we did not observe any differences by eye in relation to the dried material obtained by each of the drying methods. However, there are a number of scientific resources describing the changes that occur after drying; these include both physical and chemical modifications in the plant material [18,19]. Under the influence of temperature and dehydration, the mechanical properties, shape, size and color of the plant are modified. Thermal and gradient stresses related to water loss cause shrinkage and deformation of the dried material. At the microscopic level, internal cracks can be observed, which are the result of shrinkage stresses that tear the tissue apart [18]. In the research on dried apples, the microscopic images show cell-separated gaps; it was also shown that the volume of the created dark voids between cells increases significantly with the increase of the drying temperature [19]. The microstructure formed during the drying of plant material does not only affect its physical properties but also a number of chemical processes taking place and the rate of loss of volatile components. The decomposition reactions that lead to the formation of new compounds can take place both under the influence of enzymes and without the participation of enzymes (non-enzymatic browning). Non-enzymatic reactions are mainly related to oxidation and the Maillard reaction. At higher temperatures, the number of products formed as a result of non-enzymatic browning increases. Although there is no information available in the scientific literature on the effect of St. John's wort cellular enzymes on the drying process, as in the case of other herbs, it could also be significant. The speed of enzymatic decomposition reactions depends on the water content and temperature. The high moisture content of fresh plants favors these reactions [20].

The drying process can affect the antibacterial and antioxidant properties of herbs. Chua et al. mention that the activity of polyphenol oxidase (PPO) and peroxidase in herbs is of key importance for the preservation of phenolic and antioxidant compounds in dried herbs because these enzymes catalyze the chemical oxidation of phenolic compounds [20]. Optimum temperatures at which these enzymes retain the highest activity largely depend on the plant they come from; for most plants, it is 20–40 °C, the temperatures at which they are deactivated vary between 65 and 80 °C [21–23]. Lipases catalyze the hydrolysis of ester bonds, releasing fatty acids and organic alcohols. As Pottevin showed in 1906, in water-deficient environments, the reverse reaction (esterification) or various transesterification reactions may occur [24]. The temperatures at which these enzymes achieve the highest activity and are deactivated are similar to those of the previously mentioned enzymes (PPO and peroxidase) [25].

The temperature in the food dehydrator used for the tests was kept in the range of 44–49 °C. At elevated temperature, most likely as a result of the action of lysosomal enzymes, chemical changes occur, which result in the release of simpler compounds from cellular structures [26]. Therefore, with this drying method, much higher amounts of alcohols and short-chain fatty acids (SCFAs) were noticed. Since alcohols and acids can react with each other under these conditions, an increased number of esters has also been recorded. Most likely, higher fatty acids were also released, which, due to their low volatility, were not detected by the HS-SPME technique, whereas methyl and ethyl esters of these acids appeared among the volatile components of the dried herb. The formation of high-chain fatty acid esters is dependent on the drying temperature. These compounds were detected in all parts of the plant dried at the highest temperature (method 3). Only trace amounts of hexadecenoic acid methyl ester were found at lower drying temperatures, while no higher fatty acid esters were detected in the fresh plant.

Currently, most of the research on St. John's wort is focused on the healing properties of hypericin and hyperforin, but many studies have shown that essential oils present in St. John's wort have antioxidant, antibacterial, antifungal and larvicidal properties [27]. These essential oils can be a valuable raw material for the pharmaceutical and cosmetic industries [28]. The oils in St. John's wort are produced in the translucent canals and released via the secretory canals. These organelles are located on leaves, petals, sepals and pistils [29]. In the studied herb, the compounds from the group of monoterpenes (β-cis-

and β-trans ocimene, α- and β-pinene, γ-terpinene and limonene) and sesquiterpenes (β-caryophyllen, β-famesene, alloaromadendrene, γ-muurolene, δ-cadinene, cedrene) were found in the highest amounts. The optimal drying method should not result in a significant loss of essential oils. St. John's wort dried in a food dehydrator (method 3), compared to other drying methods, retained compounds from the group of mono- and sesquiterpenes, aromatic monoterpenes, aromatic monoterpenoids, sesquiterpenoids and aromatic sesquiterpenes in similar or slightly smaller amounts. The lowest losses when using this drying method have been observed among monoterpenoids.

Among the volatile products of St. John's wort, alkanes (20–25%) are one of the most numerous groups. This group is dominated by n-alkanes and branched alkanes with an odd number of carbon atoms. The health effects of these compounds are not clear and have not been well-studied. Herb dried in a food dryer contained significantly less of them compared to the herb dried using other methods.

### 4. Conclusions

Many herbs are used as spices, which in turn exposes them to much higher temperatures when cooked, fried or baked. Therefore, it is unlikely that harmful compounds form when drying at temperatures which do not exceed 50 °C. However, considering that herbs are a medicinal product, when using accelerated drying procedures, great care should be taken not to cause degradation or excessive evaporation of the active ingredients. Food dehydrators can be a convenient alternative to drying herbs at home, but due to the different profile of volatile components depending on the drying method, it is important to consider the amount of biologically active substances. By using various drying methods, the therapeutic effects of the herbs can be enhanced or weakened. In addition, by analyzing St. John's wort, we discovered that, at higher temperatures, new compounds are formed, most likely related to the processes of decomposition, enzymatic reactions, oxidation, mutual interactions (Maillard reaction) or thermal degradation. Depending on the humidity and temperature, these reactions take place in different proportions.

**Supplementary Materials:** The following supporting information can be downloaded at: https://www.mdpi.com/article/10.3390/pr10122593/s1, Figure S1: Chromatograms of volatile compounds extracted on PDMS/DVB fiber from flowers of St. John's wort dried by Method 1 (M1), Method 2 (M2) and Method 3 (M3). For peak designation see Table 1; Figure S2: Chromatograms of volatile compounds extracted on PDMS/DVB fiber from leaves of St. John's wort dried by Method 1 (M1), Method 2 (M2) and Method 3 (M3). For peak designation see Table 1; Figure S3: Chromatograms of volatile compounds extracted on PDMS/DVB fiber from stems of St. John's wort dried by Method 1 (M1), Method 2 (M2) and Method 3 (M3). For peak designation see Table 1.

**Author Contributions:** Conceptualization, K.D., M.J.P., S.K., M.K., B.B.-F., S.W. and A.D.-L.; Methodology, S.K. and A.D.-L.; investigation, K.D., M.J.P., S.K., M.K. and A.D.-L.; resources, M.J.P. and S.W.; writing—original draft preparation, K.D., M.J.P., S.K. and M.K.; writing—review and editing, M.K., B.B.-F., S.W. and A.D.-L.; visualization, S.K.; supervision, B.B.-F.; project administration, A.D.-L. All authors have read and agreed to the published version of the manuscript.

**Funding:** This research received no external funding and the APC was funded by the Institute of Monastery Medicine.

**Conflicts of Interest:** The authors declare no conflict of interest.

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
