# Peer review of "Influence of the Drying Method on the Volatile Component Profile of Hypericum perforatum Herb: A HS-SPME-GC/MS Study"

_processes, doi:10.3390/pr10122593_

Round 1
Reviewer 1 Report
This manuscript describes the different volatile component profiles of Hypericum perforatum L. depending on the drying method.
I found the work interesting, but scarce. I have some comments for the paper, which must be taken into account to improve the article.
Authors should include the name of the specie in italics (lines 2, 13), as well as m/z (line 101).
- The first time that an acronym is mentioned, it should be fully written. For example, HS-SPME, GC/MC, PDMS/DVB.
- Introduction: the authors should discuss more the bioactive volatile substances responsible for the medicinal effects, drying methods, etc.
- Please include references for the information exposed in lines 66-69.
- Section 2.1: more physicochemical description of the plant material is needed. For example, in lines 147-147, the authors explain that the moisture content is a factor that affects the reactions, and in lines 136-141, the authors talk about the cell-separated gaps and dark voids by microscopy. Including this type of information about the samples of the present study would be interesting.
- Section 2.2: indicate fiber conditioning parameters and the temperature during the 30 min that the fiber was inserted in the vials.
- Section 2.3: authors should explain the selection of the conditions used in the methods because the number of days and temperatures are different in all cases.
- Section 3: the information of lines 122-125 is repeated. The authors use the word “quantitative changes” (line 130-131), but they did quantify nothing. Maybe it is better to use the word “semi-quantification”.
- If the authors have some reagents, a positive confirmation with the analytical standard would be interesting.
- Authors should include the GC-MS chromatogram of some samples.
- Line 155-156: maybe it would be interesting to include complementary techniques in this study such as the determination of higher fatty acids.
- The authors found in total more than 100 volatile compounds. However, they do not discuss most of them in the results section. Differences between the number of compounds found in the different parts of the plant were not discussed. An explanation of the meaning of the values obtained for the RI 5MS and RI NIST should be included.
- Conclusions: the authors talk about how harmful some compounds can be (released at higher temperatures) and that care should be taken, but they did not discuss the toxicity of the compounds found in the present study. Maybe, it could be interesting to include the estimation of the toxicity. For example, the Cramer decision tree estimates the toxicity based on the molecular structure of the compound.
Reviewer 2 Report
-In lines 132-138, authors mention that there were changes in mechanical properties, shape, size and color, during drying, however where are these results?, is there any supplementary material section?, they mention changes observed by microscopic images, but they are not included.
-In lines 141-142 Author mention that some compounds could be produced by the action of enzymes, which?, if different enzymatic reactions were not determined, at least authors must mention according to scientific literature which could be involved?
Reference 18 has other format
Round 2
Reviewer 1 Report
The manuscript was revised very thoroughly and questions and suggestions of the reviewer were addressed. Text was changed and expanded. Due to the revision, the text is much more detailed and easier to understand. In my opinion, the manuscript is ready for publication.
Reviewer 2 Report
The suggestions were attended